# Inflammatory Bowel Disease-Associated Colorectal Cancer: Translational and Transformational Risks Posed by Exogenous Free Hemoglobin Alpha Chain, a By-Product of Extravasated Erythrocyte Macrophage Erythrophagocytosis

**DOI:** 10.3390/medicina59071254

**Published:** 2023-07-06

**Authors:** Maya A. Bragg, Williams A. Breaux, Amosy E. M’Koma

**Affiliations:** School of Medicine, Division of Biomedical Sciences, Meharry Medical College, Nashville, TN 37208, USA; mbragg20@mail.mmc.edu (M.A.B.); wbreaux20@mail.mmc.edu (W.A.B.)

**Keywords:** inflammatory bowel disease, colitis-associated colorectal cancer, exogenous free hemoglobin alpha chain, Fenton Reaction, DNA damage, haptoglobin, deferoxamine, flavonoids, hydrogen peroxide, hygiene, iron, nanomedicine, oxidative stress, polyphenol, pharmaceutical therapy

## Abstract

Colonic inflammatory bowel disease (IBD) encompasses ulcerative colitis (UC) and Crohn’s colitis (CC). Patients with IBD are at increased risk for colitis-associated colorectal cancer (CACRC) compared to the general population. CACRC is preceded by IBD, characterized by highly heterogenous, pharmacologically incurable, pertinacious, worsening, and immune-mediated inflammatory pathologies of the colon and rectum. The molecular and immunological basis of CACRC is highly correlated with the duration and severity of inflammation, which is influenced by the exogenous free hemoglobin alpha chain (HbαC), a byproduct of infiltrating immune cells; extravasated erythrocytes; and macrophage erythrophagocytosis. The exogenous free HbαC prompts oxygen free radical-arbitrated DNA damage (DNAD) through increased cellular reactive oxygen species (ROS), which is exacerbated by decreased tissue antioxidant defenses. Mitigation of the Fenton Reaction via pharmaceutical therapy would attenuate ROS, promote apoptosis and DNAD repair, and subsequently prevent the incidence of CACRC. Three pharmaceutical options that attenuate hemoglobin toxicity include haptoglobin, deferoxamine, and flavonoids (vitamins C/E). Haptoglobin’s clearance rate from plasma is inversely correlated with its size; the smaller the size, the faster the clearance. Thus, the administration of Hp1-1 may prove to be beneficial. Further, deferoxamine’s hydrophilic structure limits its ability to cross cell membranes. Finally, the effectiveness of flavonoids, natural herb antioxidants, is associated with the high reactivity of hydroxyl substituents. Multiple analyses are currently underway to assess the clinical context of CACRC and outline the molecular basis of HbαC-induced ROS pathogenesis by exposing colonocytes and/or colonoids to HbαC. The molecular immunopathogenesis pathways of CACRC herein reviewed are broadly still not well understood. Therefore, this timely review outlines the molecular and immunological basis of disease pathogenesis and pharmaceutical intervention as a protective measure for CACRC.

## 1. Core Message

Inflammatory bowel disease-associated colorectal cancer (CACRC) is becoming more prevalent worldwide and presents at a younger age. IBD, as well as CACRC, is evolving worldwide, especially in newly industrialized countries. With an aging population, its compound prevalence suggests that CACRC could become an emerging global challenge. Although surveillance and chemoprevention for CACRC exist, sixty percent of patients with CACRC are asymptomatic upon detection and over fifty percent present with advanced disease; this eventually leads to less favorable outcomes compared to sporadic colorectal cancer (SCRC). To understand why, scientists profiled surgical pathology resections of colonic mucosal and submucosal layers from patients with IBD who had undergone pouch surgery, restorative proctocolectomy with ileal pouch–anal anastomosis (RPC-IPAA) [1]. A pool of exogenous/free hemoglobin alpha chains (HbαCs) in areas of active colitis was unexpectedly found. Furthermore, the HbαCs were produced through the action of immune infiltrating cells (macrophages) that promoted reactive oxygen species (ROS) in epithelial cells depleted of colonic tissue homogenate antioxidants (i.e., nuclear factor erythroid 2-related factor 2 (Nrf2), catalase (CAT) superoxide dismutase (SOD), and glutathione peroxide (GPx)). The antioxidants above are significant regulators of cytoprotective responses to oxidative stress and the primary cellular defense against cytotoxic effects of oxidative stress [2,3,4,5,6]. Intestinal mucosal damage in IBD involves reactive oxygen metabolites (ROMs). Endogenous antioxidant enzymes neutralize ROMs in a carefully balanced two-step pathway. First, SOD converts superoxide anion to hydrogen peroxide (H(2)O(2)). Then, hydrogen peroxide is neutralized to water by CAT or glutathione peroxidase (GPO) [1]. This indicates that exogenous/free HbαC has a physiological role in inducing ROS formation and DNAD and, if not attenuated, can trigger carcinogenesis [1]. Our central focus is on the fact that HbαC induces oxygen free radical-mediated DNAD through increased ROS and decreased antioxidant defenses [1,7]. If the Fenton Reaction was mitigated by pharmaceutical therapy using haptoglobin, deferoxamine, and/or flavonoids, then this would reduce ROS, promote apoptosis and DNAD repair, and prevent the incidence of CACRC [8].

## 2. Introduction

Colorectal cancer (CRC) is often described as the “disease no one has to die from”, but approximately 50% of patients with CRC who undergo potentially curative surgery ultimately relapse and die, usually as a consequence of metastatic disease [9,10]. According to GLOBOCAN 2018 data, and the American Cancer Society, for both men and women in the United States of America, colorectal cancer (CRC) is the third main cause of cancer-related mortality in the world [11,12]. CRC is the deadliest cancer [13,14]. IBD is a known risk factor for developing CACRC [15]. IBD patients are at increased risk of CACRC due to long-standing chronic inflammation, genetic alterations, and epigenetic environmental factors [16,17,18]. Additionally, data indicate that CACRC may have evolved through a pathway of tumorigenesis distinct from that of SCRC.

Predominantly colonic IBD, the “colitides”, includes ulcerative colitis (UC) and Crohn’s colitis (CC), which are two heterogeneous, chronic relapsing and remitting gastrointestinal tract disorders in the colon [18,19,20,21,22]. Currently, both diseases affect approximately three million people in the United States. However, the incidence and prevalence of both are increasing worldwide, thus making them global emergent diseases with significant clinical challenges [22]. The global prevalence of IBD is currently evolving, approaching 90 cases/100,000 people [23], though awareness should be assessed in each of the geographical locations of the world [24,25]. North America and Canada have the highest rates of IBD in the world [26,27]. However, over the past three decades, the incidence of IBD in low-income countries has steadily risen. [26,28,29,30,31,32,33]. The burden/implication of IBD is discrete in various countries and locations, especially when contrasted/matched between low-income [34,35,36,37,38,39,40,41,42,43,44,45,46,47,48,49] and wealthy nations [50,51]. The estimated data suggest that 25 to 30 percent of cases of CD and 20 percent of patients with UC present during adolescence and young adulthood at the reproductive age [52,53,54,55,56,57,58,59,60,61]. The extent of racial/ethnic and regional differences in the prevalence of IBD in the United States remains largely unknown, warranting additional research [62,63]. However, IBD has predominantly affected white populations, particularly Ashkenazi Jews. But over the last three decades, IBD has “emerged” in minority communities [26,63,64,65,66,67,68]. The genesis of IBD is unknown, but is believed to be multifactorial [18,30,69,70]. It has been hypothesized that intestinal damage in UC and in CC is related both to increased oxygen-derived free radical production, mainly resulting from a respiratory burst of infiltrating phagocyte cells, and to a low concentration of endogenous antioxidant defense mechanisms. Indeed, neutrophils and monocytes in patients with active and/or fulminant IBD exhibit higher concentrations of oxygen-derived free radicals than in normal control samples [70,71,72,73]. Compared to other tissues, the gut is potentially more susceptible to oxidant injury/trauma, which can be exacerbated by the low concentration of antioxidant enzymes in epithelial cells, which contributes to the ROS cytotoxicity observed in the colons of patients with IBD [1,74]. IBD has no curative drug, often resulting in significant long-term comorbidity (1). The development of potential immunosuppressive therapies in IBD aims to achieve long, deep remission, but their effects on subsequent CACRC have yet to be established. However, studies have shown that the longer a person has had IBD, the higher their chance of developing CACRC [75,76,77]. An extensively referenced comprehensive meta-analysis of 19 longitudinal and cross-sectional studies with age-stratified data reported that the cumulative incidence of CACRC in UC is 2% after 10 years, 8% after 20 years, and 18% after 30 years of disease [78]. In contrast, other studies reported lower incidence rates accredited to, among other factors, the benefits of endoscopic monitoring surveillance and anti-inflammatory pharmaceutical chemoprophylaxis [79,80,81]. The greatest hope and assurance for cancer prevention in IBD depends to a large extent on broadening our, thus far, insufficient understanding of the molecular pathogenesis link between neoplastic and chronic inflammation pathways. The discovery of exogenous/free HbαC in IBD, produced through the action of immune infiltrating cells and resultant ROS production in epithelial cells, is innovative [1]. In this review, we summarize the current knowledge and awareness of CACRC genesis, focusing on the fundamental mechanism underlying its pathogenesis, and on the potential implications of the “colonic deposition of exogenous/free HbαC”, a previously unknown tissue by-product in IBD, as a possible major trigger of CACRC. Herein we discuss the “Fenton Reaction” and how exogenous HbαC could be chelated by pharmaceutical intervention to stop ROS production and promote apoptosis and DNAD repair to prevent the incidence of CACRC carcinogenesis.

## 3. People with Inflammatory Bowel Disease Are at Escalated Risk of Colitis-Associated Colorectal Cancer with a Subsequent Poor Prognosis

People who suffer from colonic IBD are at increased risk for developing CACRC [79,82]. All instances of CACRC are located in segments with colitis [75]. CACRC is one of the most severe complications of IBD, with a mortality rate of 10–15%, and the risk is 1.5–2.4-fold that in the general population [15,83]. The dysplasia of CACRC develops via a different pathway and mechanism in comparison to SCRC [15]. The well-established risk factors for CACRC are time scale and the extent of intestinal inflammatory lesions [15,75,84,85,86]. Genetic factors, coupled with the longevity of the persistent fulminant interdependent inflammatory process in the colonic mucosal layers, are believed to play a remarkable role in CACRC carcinogenesis, and consequently, inflammatory action could decrease this continuous process of inflammation associated with carcinogenesis [87,88,89]. Survivability depends on adherence to colonoscopic surveillance, and early elective colectomy is recommended [75,90,91]. However, some oncologic analyses provide positive results after curative surgeries in patients with CACRC [89,92]. This warrants continuous surveillance to assess postcolectomy safety [75,90,93].

The prevalence of CACRC development is identical for patients with UC and CC [94,95,96,97], as is the quantitative exogenous HbαC between the two colitides [1]. This timely review was conducted to summarize and determine the efficacy and pharmaceutical safety of Fenton Reaction mitigation as a preventive measure for CACRC.

## 4. Malfunctioning Tight Junction Protein CALUDIN-1 Is a Source Point of Colitis-Associated Colorectal Cancer Carcinogenesis

The tight junction is an intricate intercellular junction found in epithelial and endothelial cells that is accountable for the genesis of functional epithelial and endothelial barriers that synchronize the passage of cells and solutes through the paracellular space [98]. Patients with IBD are known to have dysfunctional claudin-1, an intestinal epithelial tight junction protein (Figure 1) [99,100]. Irregular functions in claudin-1 leads to changes in cell permeability, causing blood capillary extravasation (hemorrhage), macrophage erythrophagocytosis, and the subsequent release of free HbαC exogenously into the interstitial space, Figure 2 [1]. Within the interstitial space, HbαC is observed to serve as a biological substrate in the Fenton Reaction, producing hydroxyl radicals, as shown in Figure 3, which leads to DNA damage (Figure 4) within normal intestinal mucosa and subsequent tumor formation if the damaged DNA is irreparable [8]. This unveiled molecular understanding of chronic inflammation in patients suffering from IBD provides insight into the evolution of CACRC. Inflammation can induce mutagenesis, and the relapsing–remitting nature of this inflammation, coupled with epithelial regeneration, may exert selective pressure, accelerating carcinogenesis [101]. In summary, the sequential molecular pathogenesis of CACRC is due to inflammation, claudin-1 dysfunction, the extravasation of erythrocytes, macrophage erythrophagocytosis, and exogenous HbαC-ROS-DNAD carcinogenesis [13,47]. Within the interstitial space, HbαC acts as a substrate in the Fenton Reaction (Fe^2+^ + H_2_O_2_ → Fe^3+^ + ·OH + OH^-^) (Figure 3) [48]. The production of hydroxyl radicals in the Fenton Reaction, as shown in Figure 3, can lead to DNAD within normal intestinal mucosa and subsequent tumor formation if the damaged DNA is not repaired.

## 5. Pharmacological Mitigation of Fenton Reaction to Prevent Colitis-Associated Colorectal Cancer Oncogenesis

Ex vivo studies demonstrated a pool of free HbαCs (until recently, an unknown tissue by-product) in IBD patient mucosal microenvironments modulated by extravasated microphage erythrophagocytosis, Figure 2 [1]. In vitro data show that HbαC induced high levels of ROS production that caused DNAD, which was exacerbated by systemic decreased antioxidant defenses [1,103,104]. The focus of this study is on the fact that if the Fenton Reaction (Figure 3) were mitigated via pharmaceutical therapy, then this would reduce ROS and promote DNAD repair and apoptosis, which could prevent the incidence of CACRC [8].

## 6. Pharmaceutical Approach to Preventing Colitis-Associated Colorectal Cancer

Colonoscopy surveillance serves as the gold standard for prevention, but it has proven relatively inadequate for ascertaining the earliest molecular pathogenic relationship between neoplasia and chronic inflammation (more specifically, Fenton chemistry and its relationship with exogenous/free HbαC, hydroxyl radical (·OH) formation via the Fenton Reaction (Fe^2+^ + H_2_O_2_ → Fe^3+^ + ·OH + OH^−^), DNA damage (DNAD), and subsequent tumor formation). The Meharry-Vanderbilt alliance focuses on understanding iron chelation therapy for mitigating in vitro Fenton Reactions through a pharmaceutical approach. HbαC removal may be executed and accomplished using chelation therapy with chelating drugs, i.e., deferoxamine (DF), deferiprone (L1), and flavonoids [105,106], to attenuate HbαC toxicity.

### 6.1. Haptoglobin (Hp)

Free haptoglobin is removed from plasma in 3.5–5 days. On the other hand, the haptoglobin–hemoglobin (Hp-Hb) complex is removed within 20 min. This known fact stresses the importance of Hb removal in the presence of Hp. Haptoglobin is a tetrameric protein, a polymer built of four monomer units that contains two light (α) and two heavy (β) chains covalently bound to each other via disulfide bridges. There are three Hp phenotypes: Hp1-1, Hp2-1, and Hp2-2. Haptoglobin polymorphism occurs due to variations in the α-chain; the α-1 chain carries 83 amino acids and the α-2 chain accommodates 142 amino acids. The β-chain encompasses 245 amino acids and is not polymorphic. As shown in Figure 5, Hp1-1 is the smallest haptoglobin protein structure [107,108,109]. Further research has proven that the ability of Hp to avoid damage inflicted by free radicals is largely phenotype-pendent. Various phenotypes have the same binding affinities, but the removal of Hp from the extravascular space is size-dependent and removal of the Hp1-1:Hb complex occurs more rapidly, while the Hp2-2:Hb complex is the largest and its removal occurs more slowly. Thus, when complexed with Hp2-2, Hb-α stays in the circulation predominantly and causes enormous oxidative stress via Fenton chemistry [8,110]. Additionally, the prevalence of Hp2 is higher in IBD patients, thus contributing to reduced anti-inflammatory effects and an increased risk of CACRC development in this population [7,111].

### 6.2. Deferoxamine (DFO)

Deferoxamine (DFO) is a hydrophilic iron-chelating agent that has been shown to inhibit free radical formation [112,113] and polymeric DFO for enhancing iron chelation cancer therapy. However, as shown in Figure 6, its hydrophilic properties limit its ability to cross cell membranes and remain effective in vivo. This feature alone requires higher concentrations and longer incubation periods of DFO in order to yield anti-inflammatory effects (inhibiting the Fe-dependent production of hydroxyl radicals) from the agent. Chelation therapy would remove excess exogenous iron from the body and prevent the production of hydroxyl radicals (−111). Further, antioxidants may also play an important role. Administering antioxidants would neutralize the free radicals and block their harmful effects on intestinal cells. Salicylaldehyde isonicotinoyl hydrazone (SIH) is a lipophilic iron-chelating agent that crosses cell membranes more effectively when compared to DFO, thus requiring lower concentrations and incubation periods to produce similar anti-inflammatory effects when compared to DFO.

### 6.3. Flavonoids

Flavonoids are free radical scavengers and confer a wide variety of antioxidant and anti-inflammatory activities, as depicted in Figure 7 [115]. Studies have shown that the enteroendocrine system is composed of enteroendocrine cells (EECs) that regulate IBD by monitoring the gut microbiota and controlling the immune response, thus safeguarding the intestines against physical obstacles, as well as modulating gut motility [116]. Flavonoids have an impact on the enteroendocrine system and safeguard it against IBD, which infers that the alleviation of IBD is possibly associated with the regulation of flavonoids in EECs. Presently, over 4000 multifarious flavonoids have been recognized and ascertained in the bright colors of many fruits and vegetables [117,118]. Further, a number of studies have reported the effect of flavonoids on enterohormone secretion; however, there are hardly any studies demonstrating the association between flavonoids, enterohormone secretion, and IBD. The interplay between flavonoids, enterohormones, and IBD is herein illuminated in this review. Furthermore, the conclusion can be drawn that flavonoids may safeguard against IBD by regulating enterohormones, such as glucagon-like peptide 1 (GLP-1), GLP-2, dipeptidyl peptidase-4 inhibitors (DPP-4 inhibitors), ghrelin, and cholecystokinin (CCK), a possible mechanism of flavonoids protecting/ shielding against IBD [119].

The most likely way to reduce the incidence of oncological transformation related to IBD is via the clearance of excess exogenous HbαC from the interstitial space (Figure 8, Point D). However, this method remains limited until the malfunctioning claudin-1 (Figure 8, Point C) in the extracellular matrix in the epithelial endothelium and connective tissue is resolved to prevent petechial hemorrhage. This would be the most solid preventive measure to circumvent CACRC development.

## 7. Closing Remarks

To date, there is still no consensus on colonoscopy surveillance for patients, and it has been mentioned that few gastroenterologists adhere to the recommended number of biopsy samplings during the procedure. This further proves the point that today’s current endoscopic surveillance is inadequate, and re-emphasizes the need to look further into the dysfunctional claudin-1 protein; this could hopefully prevent ROS-mediated DNAD and the future need for colonoscopy surveillance, which has proven to be inadequate for many patients.

Supporting clinicians, in their adoption of new screening guidance for colorectal cancer by establishing and fortifying key learning approaches, may be expected to change their methods as additional research becomes available. The United States Preventive Services Task Force (USPSTF) guidelines recommend that the 45–49-year-old cohort begin screening [121,122,123]. This enforcement may help identify high-risk populations in primary care settings. Considerations for individuals at the highest risk of poor outcomes due to social determinants of health should be made, and organized screening programs should be established to eliminate barriers to care [124,125,126].

Since there is still no known cure for IBD, knowing all the factors that might worsen these diseases is quite important in order to understand and prevent disease and find therapies. More reliable biomarkers of pre-malignancy are required. Such biomarkers should help identify patients who are at increased risk of developing CACRC, and these patients should undergo personalized surveillance and treatment. Enhanced detection, particularly the removal of precancerous polyps and dysplasia, and advances in treatment have improved CRC outcomes [127,128]. The standard of care for CRC surveillance involves screening starting at age 45 for patients at average risk, and earlier, more frequent monitoring for patients with a family history of CRC. Racial minorities, however, receive unequal CRC care, and thus, experience higher incidence and mortality. African Americans (AAs) are less likely to be given a screening recommendation by their providers [129]. Likewise, a study of 5793 patients found that AAs are more likely than White Americans (WAs) to report physician non-recommendations as the predominant deterrent to screening (adjusted odds ratio of 1.46) [130]. Patient education, assistance with appointments, as well as the enhancement of physician communication and cultural competency have been shown to improve CRC screening in minorities [131,132,133]. Initiating race-specific clinical guidelines for CACRC screening in AA is needed. The implementation of pre-clinical patient navigation and fecal immunochemical testing in the community may increase CRC screening within this population. First, we need to consult the literature on disparities in CRC prevention, detection, and treatment among AAs. Next, we must develop clinical guidelines that promote CRC screening in AAs and address patient–physician communication and health literacy. Finally, we need to investigate and understand colitis–cancer sequences and their role in reducing the burden of CACRC.

## 8. Discussion

The primary causative factor for CACRC risk is thought to be a chronic inflammatory condition of the colon and rectum [134,135,136]. CACRC for UC (1925) [137] and CC (1948) [138] is a leading cause of long-term mortality. The prevalence of colorectal cancer development risk in patients with UC and CC is exactly the same [94]. Recent studies have reported that IBD confers a higher risk of CRC in males compared to females [82,139] and affects mostly middle-aged individuals [139,140]. For almost 30 years, attempts at cancer prevention have been reliant on an observational strategy of endoscopic colonoscopy surveillance with biopsies to substantiate dysplasia, the earliest recognizable precursor of CRC and the most well-founded marker of impending inevitable cancer risk. Ideally, the rationale of surveillance is to permit most patients whose biopsy specimens remain dysplasia-free to avoid unnecessary colectomy surgery, while enabling those with dysplasia to undergo prophylactic removal of the colon before the development of CRC. Although validation of this action plan has been based largely on incidental evidence, surveillance has been widely accepted and widely executed as the standard of care for patients at risk of CRC [140,141]. Although it is current, endoscopic surveillance seems to be inadequate in detecting early dysplasia that precedes CACRC. The eminent undertaking for cancer prevention in IBD is based greatly on increasing our knowledge of the molecular pathogenetic association between neoplastic and chronic inflammation pathways [95,96,97,142,143].

Despite being “the disease no one has to die from,” CRC is the most deadly cancer among males in three nations and females in five countries [13,14]. Patients with IBD, which constitutes two subclasses, i.e., UC and CD, have an increased probability of developing CACRC. This is due to prolonged fulminant chronic inflammation in the colon and rectum. CACRC risk increases with pan-colitis as well as prolonged disease duration. One meta-analysis found that the prevalence of CACRC in patients with UC was 3.7% overall compared to 5.4% in patients solely with pan-colitis. Furthermore, the risk of developing CACRC was 2% at 10 years, 8% at 20 years, and 18% at 30 years, respectively [78]. Despite endoscopic surveillance and treatment, IBD-associated CACRC is frequently diagnosed at advanced stages. In a retrospective study, Averboukh et al. [144] reviewed the medical charts of CACRC patients who had undergone RPC-IPAA surgery between 1992 and 2009. From their review, they discovered that 36% of patients presented at stage III and 17% of patients presented at stage IV, thus contributing to the poor prognosis as well as 15% of all IBD-related deaths [144]. Despite this information, further basic research needs to be conducted to implement and ascertain the molecular pathogenic relationship between neoplastic and chronic inflammation. Patients with IBD are known to have dysfunctional claudin-1, an intestinal epithelial tight junction protein (Figure 1) [99]. Irregularity in claudin-1 can lead to multifunctional cell capillary/vascular permeability, causing blood extravasation, macrophage erythrophagocytosis, and the release of exogenous/free HbαC into the interstitial space (Figure 2) [1]. Within the interstitial space, HbαC is observed to serve as a biological substrate in the Fenton Reaction (Figure 3) [102,114,120,145,146,147,148,149]. The excessive production of hydroxyl radicals in the Fenton Reaction, as shown in Figure 3, can lead to DNA damage within normal intestinal mucosa and subsequent tumor formation if the damaged DNA is not repaired.

## 9. Significance

To date, there is no pharmaceutical cure for IBD. Knowing all the factors that might worsen these diseases is quite important to understand symptomatology management. More reliable biomarkers of pre-malignancy are needed to help recognize patients who are at increased risk of developing CACRC and to select such patients for personalized surveillance, management, and treatment. According to the American Cancer Society, in the United Statesm CRC incidence has doubled in younger adults and is the third leading cause of cancer deaths. The incidence of colorectal cancer (CRC) is rapidly increasing among younger individuals, and the disease is also being diagnosed at more advanced stages at all ages, according to a new report from the American Cancer Society. Diagnoses in people younger than 55 years doubled from 11% (1 in 10) in 1995 to 20% (1 in 5) in 2019. In addition, more advanced disease is being diagnosed; the proportion of individuals of all ages presenting with advanced-stage CRC increased from 52% in the mid-2000s to 60% in 2019. The rates are increasing in young people, and it is alarming to see how fast the whole patient population is becoming younger, despite decreasing numbers in the overall population (the American Cancer Society)

Enhanced detection, particularly the removal of precancerous polyps and dysplasia, and advances in treatment have improved CRC outcomes [127,128]. The standard of care for CRC surveillance is screening starting at age 45 for patients with average risk, and earlier, more frequent screenings are performed for patients with a family history of CRC. Racial minorities, however, receive unequal CRC care, and thus, experience higher incidence and mortality. A study additionally conveyed that AAs were less likely to be given a screening recommendation by their provider [129]. Likewise, a study of 5793 patients found that AAs were more likely than White Americans (WAs) to report physician non-recommendations as the predominant deterrent to screening (adjusted odds ratio of 1.46) [130]. Patient education, assistance with appointments, as well as the enhancement of physician communication and cultural competency have been shown to improve CRC screening in minorities [131,132,133]. The implementation of pre-clinical patient navigation and fecal immunochemical testing in the community may increase CRC screening within this population. First, this requires consulting the literature on disparities in CRC prevention, detection, and treatment among AAs. Second, clinical guidelines must be developed that promote CRC screening in AAs and address patient–physician communication and health literacy. Third, we must describe colitis–cancer sequences and the mediating conditions characterizing their role in reducing the burden of CACRC incidence. Finally, IBD-related health disparities exacerbate the CACRC mortality rate. African American patients with IBD are almost twice as vulnerable to the development of CRC when compared to their White Americans (WA) counterparts. Although early screenings (i.e., endoscopic/colonoscopy surveillance) have been proven to reduce CACRC, AAs have not benefited from such preventative strategies secondary to non-compliance [14]. Thus, there is a need to generate alternative preventative measures. If mitigation of the Fenton Reaction is successful, then this would: (i) reduce the incidence of CACRC and its mortality; (ii) reduce and/or eliminate the need for endoscopic/colonoscopy screening for IBD patients, which is not favorably viewed by AAs; and (iii) eliminate non-compliance with screening, and thereby reduce CRC morbidity in AAs.

## 10. Limitations

There are neither pharmaceuticals to cure IBD nor solutions to restore and normalize the physiology of the dysfunctional tight junction of the capillary endothelial “claudin-1” during active IBD. Dysfunctional claudin-1 triggers potential hemorrhages and subsequent sequences that lead to the development of CACRA.

## 11. Ethical Considerations

This study was conducted in compliance with the ethical standards of the 1975 Declaration of Helsinki, as revised in 2008, and the European Union’s Guidelines for Good Clinical Practice [150,151]. According to the cited references disseminated in peer-reviewed journals and scientific meetings written informed consent was obtained from patients. This project was authorized by the Meharry Medical College and Vanderbilt University Medical Center Institutional Review Boards (IRB # 100916AM206, 080898, and 100581).

## Figures and Tables

**Figure 1 medicina-59-01254-f001:**
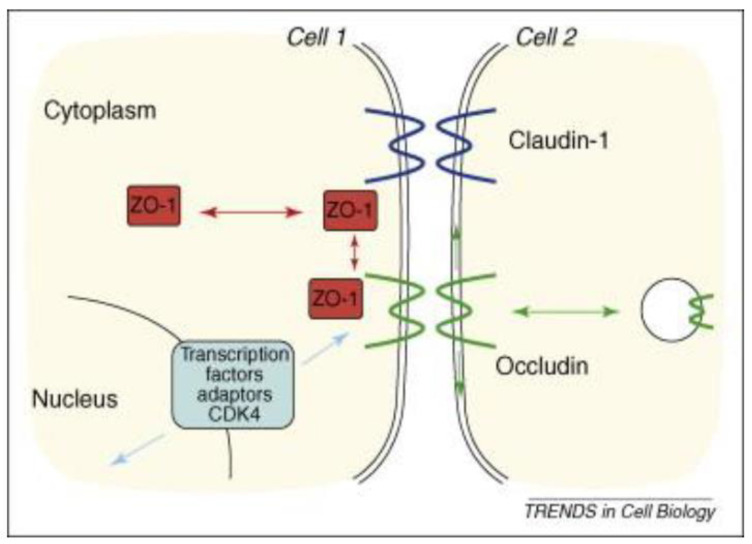
Dynamic and functional signaling pathways for tight junctions and the epithelial junction complex: The figure shows a schematic drawing of the malfunctioning tight junction protein claudin-1 due to inflammation in IBD. Tight junctions are an intercellular adhesion complex of epithelial and endothelial cells, and form a paracellular barrier that restricts the diffusion of solutes on the basis of size and charge. Tight junctions are formed of multiprotein complexes containing cytosolic and transmembrane proteins. Reproduced with permission from Steed et al., Trend Cell Biol, Elsevier, 2010 [98]. Abbreviations: ZO-1 is a tight junction protein that establishes a link between the transmembrane protein occludin and the actin cytoskeleton; occludin, is an integral transmembrane component of the tight junction.

**Figure 2 medicina-59-01254-f002:**
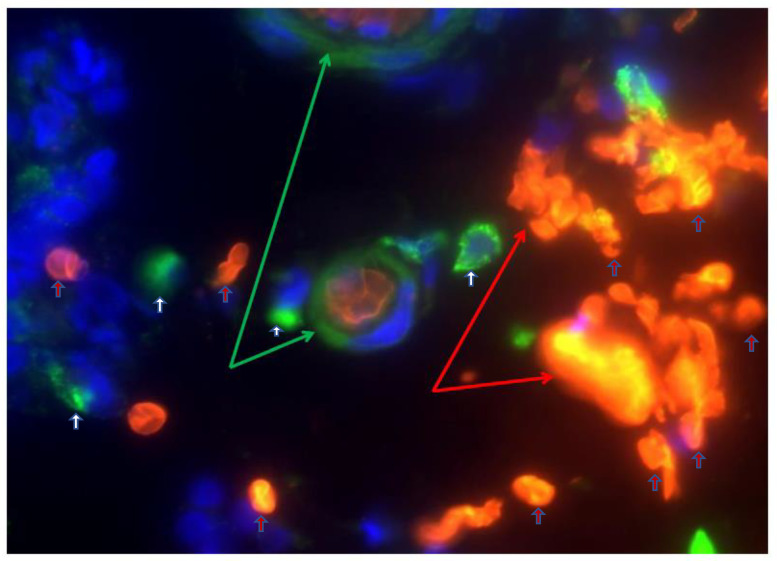
Depiction of extravasated erythrocyte macrophage erythrophagocytosis (EEME): Double immunofluorescent staining of the macrophage marker CD163 (green arrow) in paraffin-embedded sections from a patient with UC. Nuclei were counterstained with DAPI (blue). Red arrows indicate extravasated erythrocytes and white arrows indicate macrophages. Pictures were taken at 60× magnification. Green arrows depict macrophage erythrophagocytosis—a macrophage engulfing three extravasated erythrocytes (red arrows). Reproduced with permission from Myers et al., Inflamm Bowel Dis, 2014 [1]. Abbreviations: EEME, extravasated erythrocyte macrophage erythrophagocytosis; UC, ulcerative colitis.

**Figure 3 medicina-59-01254-f003:**
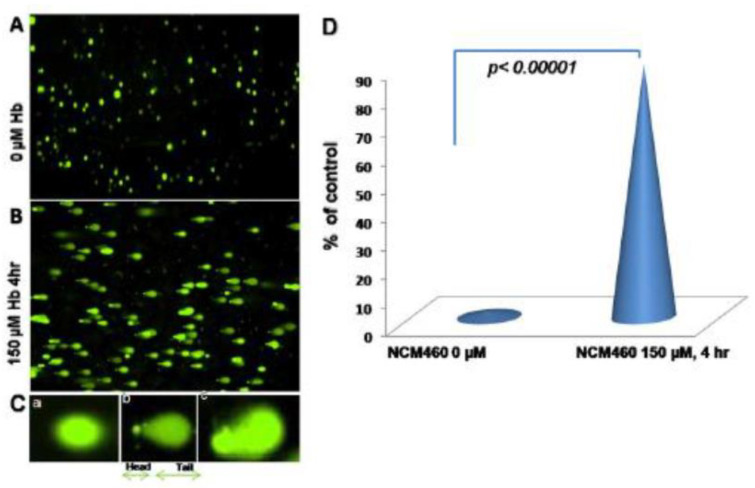
Comet assay, with Hbα causing DNA damage: (**A**) Normal human colon epithelial cells, NCM460, were fed with fresh complete media alone as a control, (**B**) with 150µM Hbα for 4 h followed by Comet assay to assess DNAD using denaturing electrophoresis. (**C**) (**a**) Undamaged cells, (**b**) damaged cells, and (**c**) severely damaged cells in an interpretation of the intensity of DNAD. (**D**) Quantification of the DNAD between the two groups. Reproduced with permission from Myers et al., Inflamm Bowel Dis, 2014 [1]. Abbreviations: NCM460, normal colonic epithelial cell line; DNAD, deoxyribonucleic acid–damage; Hbα, hemoglobin alpha.

**Figure 4 medicina-59-01254-f004:**
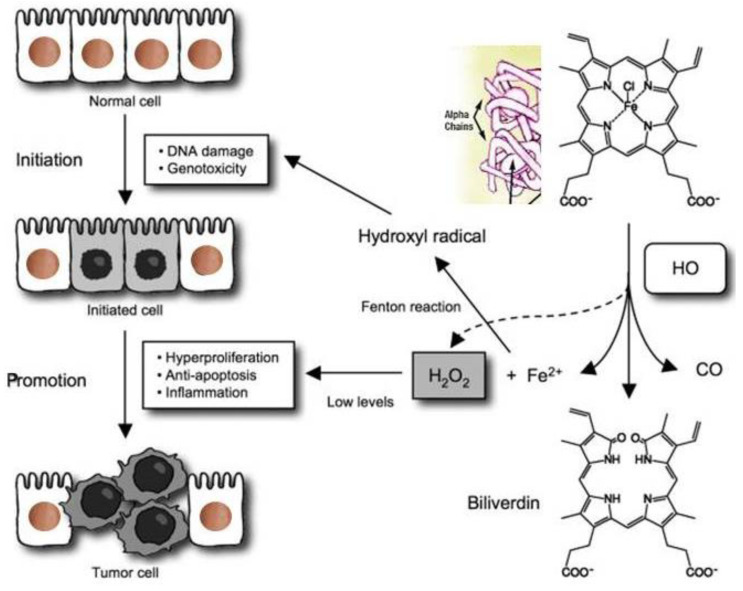
The pathophysiology of extracellular/exogenous HbαC and subsequent tumor associated with enhanced oxidative reaction, “the Fenton Reaction (FR)”: The FR here is the chemical response between exogenous HbαC and hydrogen peroxide, resulting in a hydroxyl radical, which is extremely receptive and exceedingly toxic/noxious to living cells and is an oncogenic trigger; this can also serve as a therapeutic target/strategy for cancer patients. The figure was downloaded for free, and modified for clarification [102]. Abbreviations: FR, Fenton Reaction; HbαC, hemoglobin alpha chain; HO, HO+, hydroxide, OH-, oxyhydride; CO, carbon monoxide; Fe^2^, iron (II); H_2_O_2_, hydrogen peroxide.

**Figure 5 medicina-59-01254-f005:**
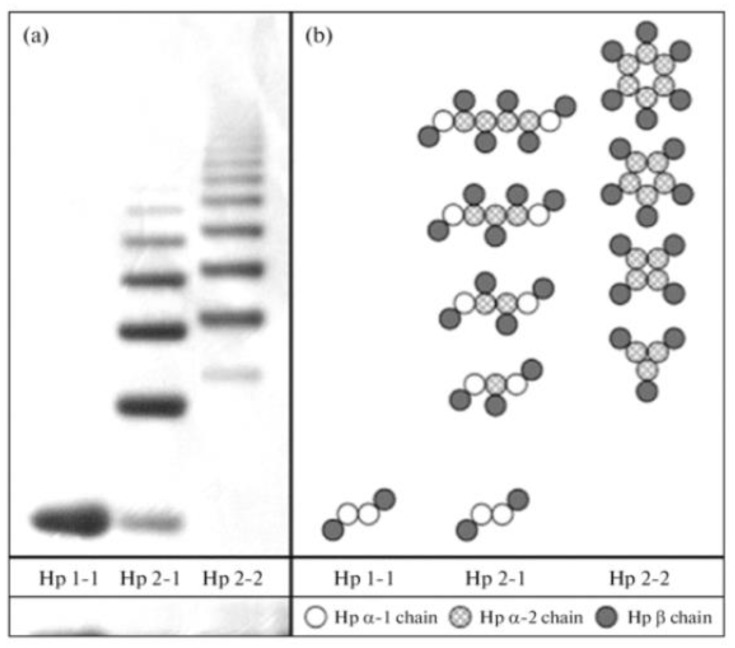
Haptoglobin phenotype evaluation via a native one-dimensional gel electrophoresis technique. Haptoglobin is the protein made in the liver that, in humans, is encoded by the HP gene. (**a**) specific profiles obtained by electrophoresis in gradient (3-8%0 native PAGE of haptoglobin preparations of various phenotypes. (**b**) Composition of polymers ot three haptoglobin phenotypes. Three major haptoglobin phenotypes are known to exist: Hp 1-1—homodimers; Hp 2-1—liner heterodimers; and Hp 2-2—cyclic heterodimers. Hp 1-1 is biologically the most effective in binding free hemoglobin and suppressing inflammatory responses associated with free hemoglobin. Hp 2-2 is biologically the least active, and Hp 2-1 is moderately active. In blood plasma, haptoglobin binds with high affinity to free hemoglobin released from erythrocytes, and thereby inhibits its deleterious oxidative activity. Free haptoglobin is removed from plasma in 3.5–5 days. On the other hand, the haptoglobin–hemoglobin (Hp-Hb) complex is removed within 20 min. This known fact stresses the importance of Hbα removal in the presence of Hp. Reproduced with permission from Naryzny et al., Biochem Mosc Suppl B Biomed Chem. Springer Nature, 2021 [107]. Abbreviations: PAGE, polyacrylamide gel electrophoresis; Hp 1-1, Hp 2-1, and Hp 2-2—Haptoglobin–hemoglobin complex, Hbα, and hemoglobin alpha.

**Figure 6 medicina-59-01254-f006:**
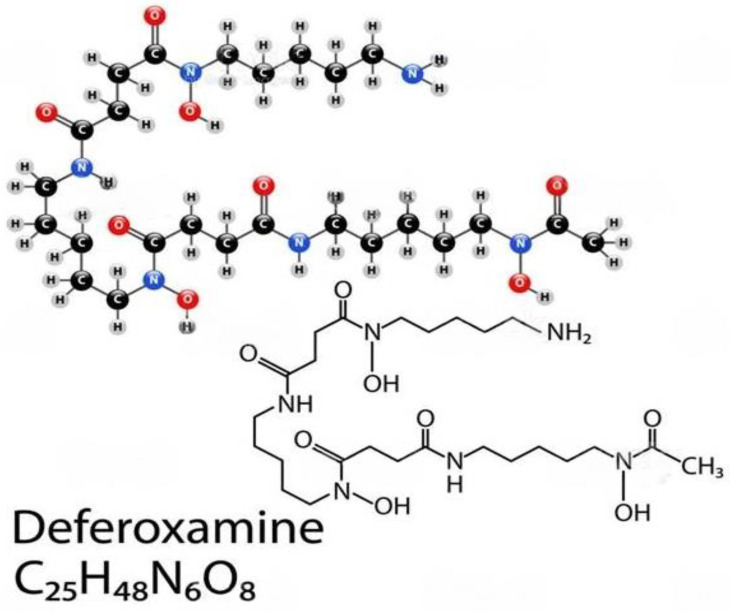
Deferoxamine, also known as desferrioxamine or desferal, is a chelating agent that is utilized to clear away unwanted excess iron or aluminum from the body. It reacts by confining exogenous free iron or aluminum in the bloodstream and reinforcing its elimination in the urine. Reproduced with permission from Cao et al., American Chemical Society, 2020 [114]. Abbreviations: C_25_H_48_N_6_O_8_, deferoxamine.

**Figure 7 medicina-59-01254-f007:**
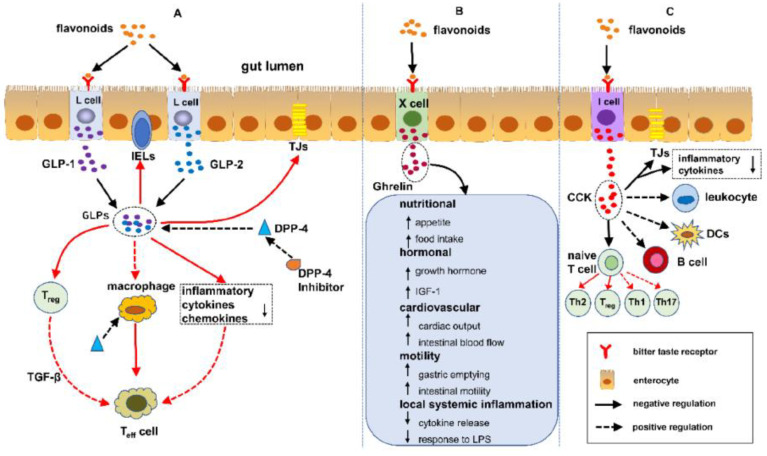
Pathway of flavonoids shielding against IBD and regulating the enteroendocrine system. (**A**) Flavonoids attune IBD via the DPP-4/GLPs pathway: (1) safeguarding the gut luminal barrier, (2) regulating Treg and intraepithelial lymphocytes (IELs) by controlling their discernment and function, and (3) modifying the task of macrophages and dendritic cells. (**B**) Flavonoids synchronize IBD via the ghrelin pathway: (1) increased food consumption, (2) increase in growth hormone action, (3) cardiovascular effects, (4) enhanced motility, and (5) reduced local and systemic inflammation. (**C**) Flavonoids control IBD via the cholecystokinin (CCK) pathway: (1) reducing the mucosal production of proinflammatory cytokines and safeguarding the intestinal barrier, (2) decreasing leukocyte migration and impending dendritic cell (DCs) activation, and (3) regulating T cells and B cells [113,115,116,117,118]. Flavonoids free haptoglobin is cleared from plasma in 3.5–5 days. On the other hand, the haptoglobin–hemoglobin (Hp-Hb) complex is removed within 20 min. This known fact stresses the importance of Hbα removal in the presence of Hp. However, this is a spinning intervention and does not solve the problem while finding a solution to dysfunctional claudin-1. Reproduced with permission from Li et al. Metabolites, published by MDPI, 2022, under the terms and conditions of the Creative Commons Attribution license [119]. Abbreviations: DPP-4/GLPs, dipeptidyl peptidase-4 (DPP-4) inhibitors block the breakdown of GLP-1 and GIP to increase the levels of active hormones; IELs, intraepithelial lymphocytes; CCK, cholecystokinin; DCs, dendritic cells.

**Figure 8 medicina-59-01254-f008:**
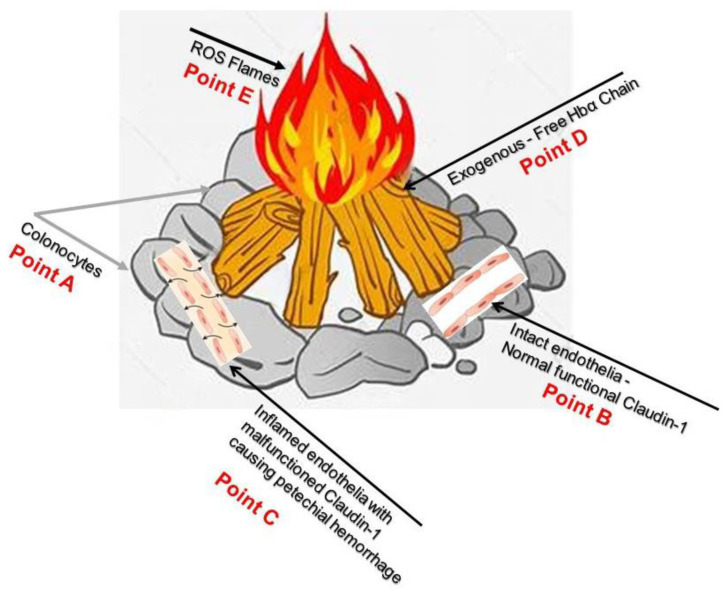
Point A demonstrates colonocytes of the colonic mucosal epithelium, while Point B illustrates intact healthy capillary endothelia with normal functional Caludi-1. Point C depicts active IBD and dysfunctional Claudin-1, the source of potential hemorrhage. Point D show exogenous free HbαCs that cause ROS flames at Point E. Current endoscopic surveillance is inadequate and re-emphasizes the need to look further into the dysfunctional claudin-1 protein; this could hopefully prevent ROS-mediated DNA damage. This figure is a visualization of the pathophysiology of CACRC (A–E). Reproduced with permission from Pinbest.com [120]. Abbreviations: HbαC, hemoglobin alpha chain; ROS, reactive oxygen species; CACRS, colitis-associated colorectal cancer.

## Data Availability

No new data were generated or analyzed in support of this research.

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
