# Peer review of "Inflammatory Bowel Disease-Associated Colorectal Cancer: Translational and Transformational Risks Posed by Exogenous Free Hemoglobin Alpha Chain, a By-Product of Extravasated Erythrocyte Macrophage Erythrophagocytosis"

_medicina, 2023, doi:10.3390/medicina59071254_

Round 1

Reviewer 1 Report (Previous Reviewer 1)

The authors carefully responded to each of the reviewers' questions and comments. 

I totally agree the authors that the problem of IBD-related colorectal cancer screening is associated with racial disparities. Again, I think the title and the content of this review (Discussion and the Significance of this paper) do not match. Is it possible to change the title of the manuscript to reflect the current issues in colorectal cancer screening?

Author Response

Question: I totally agree with the authors that the problem of IBD-related colorectal cancer screening is associated with racial disparities. Again, I think the title and the content of this review (Discussion and the Significance of this paper) do not match. Is it possible to change the title of the manuscript to reflect the current issues in colorectal cancer screening?

Response: This paper is a broadly comprehensive review. We do not know exactly what triggered IBD but is believed to be multifactorial. To understand the IBD-associated-CRC issue, which is sequentially different from sporadic CRC a basic understanding is necessary. We analyzed inflamed colon layers of UC and CC and unveiled a pool of exogenous free hemoglobin alpha chains (PMID: 25078150, reference #1) predisposed by infiltrating immune cells, extravasated erythrocytes, and macrophage erythrophagocytosis (Fig. 2) secondary to multifunctional tight junction protein “Claudin 1 (Fig. 1). These IBD patients have been shown to have increased reactive oxidative stress (ROS), DNA oxidation products, free iron in the mucosa, preneoplastic, and colitis-cancers. Our experience and that of others have clearly shown that all IBD-related CRC lesions are found in segments with colitis (references 15, 75, 84-86).

*We are requesting the reviewer’s blessing to allow us to keep the title unchanged. This is because it covers the nature presentation of the content of this paper. Thank you for your consideration.

Although “colorectal screening” is still medically a challenge, screening guidance for CRC is establishing and fortifying key learning approaches that may be expected to change as additional research becomes available. Please note even if everyone is timely screed it will not solve the IBD-CRC sequence but may improve the diagnosis rate. The major important point is to pharmacologically solve the permeability issue caused by the tight junction malfunctioning Caludin-1, the source of “petechial hemorrhage, Fig. 8 Point C (reference 149).

Reviewer 2 Report (Previous Reviewer 2)

I would like to thank the Authors for the revision which improved the manuscript.

However, the abstract did not change. Authors added one more word instead. Please, be more concrete and shorten the abstract.

Secondly, 5. Pharmacological mitigation 230 line and F 231 line did not change. This part is still not finished or should be deleted. 

Thirdly, as requested, I did not find limitations of this review.

Author Response

Reviewer 2.

Question: However, the abstract did not change. The authors added one more word instead. Please, be more concrete and shorten the abstract.

Response: We thank the reviewer for the comment and suggestion. We have revised and shortened the abstract.

Question: Secondly, 5. The pharmacological Mitigation 230 line and F 231 line did not change. This part is still not finished or should be deleted.

Response: We cannot thank the reviewer enough to have noticed this error. Thank you. We have deleted “5. Pharmacological mitigation and F”.

Question: Thirdly, as requested, I did not find limitations in this review.

Response: We truly thank the reviewer for bringing up this important part of the limitations. We have added the limitation section as # 10. Science is clear that CACRC is correlated with the exogenous free HbαC the byproduct of infiltrating immune cells, extravasated erythrocytes, and macrophage erythrophagocytosis. The limitation is technology: There are no pharmaceuticals to cure IBD nor do solutions to restore and normalize the physiology of the dysfunctional tight junction of the capillary endothelial “Claudin-1” during active IBD. Malfunctional Claudin-1 triggers potential hemorrhage and subsequent sequences to the development of CACRA.

Round 2

Reviewer 1 Report (Previous Reviewer 1)

I have no comment and question on the manuscript.

This manuscript is a resubmission of an earlier submission. The following is a list of the peer review reports and author responses from that submission.

Round 1

Reviewer 1 Report

Major comments

1)     This study reviews the exogenous free hemoglobin alpha chain as a risk factor for colon cancer in IBD. However, in the section of Discussion and Significance of this paper, problems of colorectal cancer screening and racial disparities are discussed. Therefore, it may give the impression that the title of the paper does not match the content of the paper.

2)     Some figures appear to have been reprinted from other papers, please cite references accordingly.
1. Was the image of Figure 1 reprinted from a paper somewhere? Trends in Cell Biology? Please clearly cite the reference. Please add the legends for ZO-1 and occludin.
2. Please show the experiment type of Figure 2. Immunostaining? What kind of tissue was used?
3. The legends for Figure 3(D) are missing. Please check the spelling. Comey? or Comet?
4. What does the “biliverdine” mentioned in Figure 4 mean? Please add the legends for HO.
5. Please show the experiment type of Figure 5 (a) and (b). a: image of electrophoresis?
6. Was the image of Figure 6 reprinted from a paper somewhere? Unnecessary characters can be seen in the background of this figure.

Minor comments

1)     Please check whether Deferoxamine (DFO) is hydrophilic of hydrophobic.
Line 251: Deferoxamine (DFO) is a hydrophilic iron-chelating agent…
Line 252: ….hydrophobic properties…

2)     Line 311: The underline is necessary?

3)     Line 326: Please check the patients?

4)     Please remove the extra space in the sentence. ex) Line 145, 152, etc.

5)     Please unify fonts. ex) Line 84-92.

6)     Line 405: Where does the quotation mark (“) start?

7)     The statement that "CRC is the 3rd leading cause of cancer deaths" is repeated three times in the text in the section of Introduction, Discussion, and Significance. Please summarize the epidemiological information in the background.

Minor editing of English language required.

Reviewer 2 Report

In this paper, Maya A. Bragg et al. presented a well-described review that meets the scope of the journal's special issue. Nevertheless, the Authors should address some points:

1. The current abstract version is too long (no more than 300 words maximum according to "Instructions for Authors").

2. Paragraph "Pharmacological mitigation" in lines 205 and 206 is unfinished or should be deleted.

3. Can the authors provide limitations of this review?